# Systematic Review and Meta-Analysis on the Association Between Daily Niacin Intake and Glaucoma

**DOI:** 10.3390/nu16213604

**Published:** 2024-10-23

**Authors:** Constantin Alin Nicola, Maria Cristina Marinescu, Anne Marie Firan, Mihaela Simona Naidin, Radu Constantin Ciuluvica, Maria Magdalena Rosu, Andreea-Daniela Meca, Maria Bogdan, Adina Turcu-Stiolica

**Affiliations:** 1Doctoral School, University of Medicine and Pharmacy of Craiova, 200349 Craiova, Romania; 2Discipline Physiology III, Faculty of Medicine, Carol Davila University of Medicine and Pharmacy, 020021 Bucharest, Romania; 3Barnsley Hospital, Barnsley Hospital NHS Foundation Trust, Barnsley S75 2EP, UK; 4Department of Pharmaceutical Marketing and Management, Faculty of Pharmacy, University of Medicine and Pharmacy of Craiova, 200349 Craiova, Romania; adina.turcu@umfcv.ro; 5Discipline Anatomy, Faculty of Dentistry, Carol Davila University of Medicine and Pharmacy, 020021 Bucharest, Romania; 6Department of Nutrition and Dietetics, Faculty of Midwives and Nursing, University of Medicine and Pharmacy of Craiova, 200349 Craiova, Romania; 7Department of Pharmacology, Faculty of Pharmacy, University of Medicine and Pharmacy, 200349 Craiova, Romania

**Keywords:** niacin, vitamin B3, glaucoma

## Abstract

Background: Glaucoma is a progressive optic neuropathy, characterised by a complex pathophysiology, with mitochondrial dysfunction playing a significant role in the cellular damage and apoptosis of ganglion cells. Niacin is a precursor to several molecules acting as coenzymes in the mitochondrial production of ATP, in DNA repair and in the reduction of reactive oxygen species. The objective of this systematic review is to assess the impact of daily niacin intake on glaucoma. Methods: Case–control and cohort studies regarding niacin and glaucoma, indexed in PubMed, Web of Science, Cochrane and Scopus, were included. Other study methodologies, studies regarding niacin in other ocular disease or other nutrients in glaucoma were excluded. Bias was assessed using the Newcastle–Ottawa Scale. The study protocol was registered in the PROSPERO database (no. CRD42024578889). Results: Five case–control studies were included. In the pooled analysis, a significantly higher proportion of patients with high niacin consumption was found in the group without glaucoma compared to those with glaucoma as defined by ISGEO criteria (*p*-value < 0.00001; OR = 0.66, 95% CI 0.55–0.79) or as defined by retinal imaging (*p*-value = 0.02; OR = 0.63, 95% CI 0.43–0.94). Conclusions: Daily dietary intake of niacin is significantly lower in patients with glaucoma compared to the general population. Given different average daily intakes of niacin in these populations, different glaucoma definitions and several confounding variables which weaken the associations, large sample, standardised randomised controlled trials are needed to confirm the potential benefits of niacin in glaucoma.

## 1. Introduction

Glaucoma is a progressive optic neuropathy, characterised by a specific loss of retinal nerve fibres and ganglion cells, and in which the most impactful risk factor is intraocular pressure—as it is the only one we can currently address therapeutically [1]. This disease is the main cause of irreversible vision loss worldwide, with a prevalence reported in 2017 of 75.6 cases per 100,000 people [2]. It is estimated that in 2020, 76 million people were affected, while the number is projected to grow to 111.8 million by 2040 [3].

The pathophysiology of glaucoma is complex and insufficiently understood; the main pathological step in the disease progression is the retinal ganglion cells undergoing apoptosis, and while high intraocular pressure (IOP) is the main risk factor, due to impaired aqueous humour outflow through the trabecular meshwork, there may be an important role in other factors as well [4,5,6]. Such factors include poor blood perfusion to ocular structures (a mismatch between IOP and systemic arterial pressure), a higher susceptibility of the lamina cribrosa to IOP aggression (leading to mechanical axonal damage), mitochondrial impairment or inflammatory processes [4,5,6]. Main recognised pathogenic theories include the vascular and mechanical theories [7].

The mitochondrial dysfunction hypothesis aims to target the metabolic function of retinal ganglion cells, thus being under the wider umbrella of neuroprotection. There are several molecules investigated in glaucoma for neuroprotection; however, none have yet translated to definitive clinical benefit in humans: citicoline (involved in phospholipid synthesis), pyruvate (involved in glycolysis and gluconeogenesis) and the focus of the present review—nicotinamide (involved in the NAD salvage pathway) [8]. In this review, the aim is to assess the relationship between niacin intake and the prevalence of glaucoma.

Nicotinamide is a derivative of niacin, more widely known as vitamin B3 [9]. In a normal diet, we may find several precursors of vitamin B3: the amino acid tryptophan; nicotinamide, niacin, nicotinic acid and nicotinamide riboside, which are transformed to the active form, nicotinamide adenine dinucleotide (NAD), through a complex biochemical pathway [10]. There are two biochemical pathways of NAD synthesis in the human body: de novo, from dietary tryptophan, and the salvage pathway, from nicotinamide [11]. NAD plays several essential roles in the body. It is a coenzyme in the mitochondrial production of ATP, in DNA repair, in secondary messenger signalling related to immunomodulation and inflammation and in the regulation of gene transcription, and a related molecule, NADPH, participates in the reduction of reactive oxygen species [10].

Niacin is highly involved in mitochondrial functions through its derivative, NAD, and its reduced form, NADH. NAD+ is an important cofactor of several dehydrogenase enzymes, thus being involved in the mitochondrial citrate metabolism and respiratory chain, also serving in other mitochondrial reaction chains, such as the fatty acid oxidation and pyrimidine biosynthesis [12]. Significant in neurodegeneration is NAD’s role as a cosubstrate for sirtuin deacetylases [12,13]. Sirtuin-catalysed protein deacylation is always coupled with the cleavage of NAD+, and through this enzymatic process, sirtuins modulate mitochondrial stress through protein quality control, antioxidant defence, mitochondrial membrane stabilization and the mediation of mitochondrial fusions, fissions and mitophagy [14].

It has been described that neurons are highly sensitive to NAD levels [15]. The pathway of transforming tryptophan to NAD is disrupted in diseases such as Parkinson’s and Alzheimer’s disease, thus decreasing the intracellular NAD levels [15]. Such preclinical data support the potential role of nicotinamide as a neuroprotective agent. Therefore, nicotinamide has been involved in preclinical investigations in several neurodegenerative diseases, including glaucoma; however, there are few clinical studies, and the results are differing. A systematic review involving animal models of Alzheimer’s disease showed that nicotinamide supplementation leads to improved learning and memory [16]. A phase I randomised trial involving Parkinson’s patients showed that supplementation with high-dose nicotinamide riboside was well tolerated and led to improvements in the Parkinson’s Disease Rating Scale [17]. Lastly, there is tentative proof that nicotinamide may be beneficial in muscular dystrophies; long-term supplementation of nicotinamide riboside in ataxia telangiectasia led to improvements in coordination and eye movement in a single-arm, open-label clinical trial [18].

Due to its reported neuroprotective effects, nicotinamide has been proposed as an adjuvant in glaucoma [19]. Several preclinical studies on animal models have suggested a robust mechanism of protection on the optic nerve [20,21]. A rat model of glaucoma showed that the oral administration of nicotinamide led to axonal preservation, lower ganglion cell loss and an increase of adenosine monophosphate-activated protein kinase, highly involved in neural signalling pathways [22]. In a mouse model of glaucoma, nicotinamide prevented glaucomatous neurodegeneration in 70–95% of cases; at a higher dose, it also had an IOP-lowering effect, besides the hypothesized neuroprotection [23]. Currently, research is being planned and performed on human patients with glaucoma to clarify the translation of these data into clinical practice. RNA sequencing involving enucleated human eyes showed that the neuroretinal tissue was expressed in a high degree in the NAD synthesis enzymatic pathway, and a key enzyme in this pathway was downregulated in glaucoma [24].

## 2. Methods

### 2.1. Information Sources and Search Strategy

We conducted a systematic review and meta-analysis to explore the relationship between dietary niacin intake and the prevalence of glaucoma. To ensure transparency and reduce the risk of research duplication, the study protocol was registered in the PROSPERO database (registration no. CRD42024578889). The findings of our review were reported following the PRISMA 2020 checklist (Appendix A).

We performed a comprehensive search across PubMed, Web of Science and Scopus databases from 2004 until August 2024. The search terms included: (niacin OR (vitamin B3) OR nicotinamide OR niacinamide OR (nicotinic acid amide)) AND (glaucoma OR (glaucomatous neuropathy))—as detailed in Table 1. We also reviewed the reference lists of pertinent articles to identify additional studies eligible for inclusion.

We excluded other types of articles (in vitro studies, animal model studies, reviews, systematic reviews and meta-analyses, editorials and manuscripts detailing only a study protocol with no projected results), studies on nicotinamide in other ocular pathologies (for instance, other optic neuropathies, such as Leber) and studies on other nutrient supplementation or dietary intake in glaucoma, as listed in Table 2. This selection was performed in order to select and pinpoint the effect of niacin intake on glaucoma.

### 2.2. Selection and Data Collection Process

One author (M.C.M.) identified all studies through database searches applying the previously described search filters and compiled a database using Microsoft Excel (for Microsoft 365 MSO, Version 2409). Two other authors (C.A.N. and M.S.N.) independently screened these records and assessed their eligibility. Finally, one author (A.M.F.) compared and reconciled the two lists, compiled the final list of studies included in the review and prepared the evidence summary table. If any inconsistencies arose, and to minimize the subjectivity of the selection process, one author (R.C.C.) reviewed all cases in which the two lists were in disagreement (blinded to who had created the selection) and independently reached a decision.

### 2.3. Data Items

The exposure followed by all studies has been the daily niacin intake, calculated following a 24 h recall questionnaire performed during the participants’ interview [25,26,27,28,29].

The outcome that was followed in all cross-sectional studies was the prevalence of glaucoma. The outcome of the effect measure was presented as the odds ratios (OR) with the corresponding 95% confidence intervals (95% CI). However, the outcome of glaucoma was defined in three different ways:-Self-reported glaucoma, based on the questionnaire administered in the studies [27,29].-Retinal imaging [27,28]: digital fundus photography was performed with a Canon ophthalmic digital imaging system and glaucoma specialists from the Wilmer Eye Institute at Johns Hopkins University evaluate the images and graded them as no, possible, probable, definite glaucoma or unable (to grade [27,28]). In the study by Lee et al., glaucoma diagnosis was defined by the presence of ‘definite’ glaucoma in either eye, whereas in Taechameekietichai et al., it was defined by the presence of either ‘probable’ or ‘definite’ glaucoma in either eye [27,28].-The International Society for Geographical and Epidemiological Ophthalmology (ISGEO) glaucoma criteria for population-based prevalence surveys [30] were applied in the studies by Jung et al. (2018), Taechameekietichai et al. (2021), Hou et al. (2024) and Lee et al. (2020) [25,26,27,29].


In the studies by Jung et al. and Lee et al., participants were drawn from the Korean National Health and Nutrition Examination Survey (KNHANES), a population-based, cross-sectional survey in South Korea [20,21]. Participants in Jung et al. were recruited between 2008 and 2012 (The KNHANES IV–V study phase), while the recruitment period for Lee et al. was not specified [25,26].

In the studies by Taechameekietichai et al., Lee et al. and Hou et al., participants were drawn from the 2005–2008 NHANES, a population-based study conducted in the United States [27,28,29].

### 2.4. Study Risk of Bias Assessment

As the studies we compiled followed the nutrient daily intake of patients, they were non-randomised studies; therefore, we chose the Newcastle–Ottawa Scale (NOS) for assessing the quality of nonrandomised studies [31]. The Newcastle–Ottawa Scale is an intuitive tool that assesses case–control studies based on three criteria: selection and definition of cases and controls (a maximum of 4 stars can be awarded for a study), comparability between cases and controls (a maximum of 2 stars) and exposure (a maximum of 3 stars). Two authors (C.A.N. and M.C.M.) evaluated all full text manuscripts and completed the NOS scale checklist for each manuscript. A third author (A.M.F.) compared and reconciled the two evaluations if necessary. No automation tools were used.

### 2.5. Statistical Analysis

We utilised RevMan 5.4 (The Cochrane Collaboration, Copenhagen, Denmark, 2020) for data analysis. Depending on the level of heterogeneity among the included studies, we applied either a random-effects or fixed-effects model. Heterogeneity was assessed using the Q-test, and Higgins and Thompson’s I^2^ statistic was reported. In cases where heterogeneity was detected (*p*-value < 0.05, I^2^ > 80%), a random-effects model was employed. The heterogeneity of results was visually evaluated through forest plots displaying pooled estimates. The outcomes were combined as the odds ratios (OR) with the corresponding 95% confidence intervals (95% CI). For meta-analyses involving more than five studies, the rank correlation test and regression test were conducted to assess funnel plot asymmetry, with a *p*-value < 0.05 indicating no asymmetry and no publication bias in the included studies.

## 3. Results

### 3.1. The Articles Selection Scheme

After searching for the keywords and applying the inclusion and exclusion criteria previously mentioned in the databases search, the query produced an initial list of 478 manuscripts, as shown in Figure 1. A number of 49 duplicates were removed, and after titles and abstracts were evaluated, a further 292 articles were eliminated (preclinical studies: 58 in vitro studies; 64 studies on animal models; 116 reviews and editorials; 1 study protocol, with no results regarding B3 and glaucoma; 53 other types of articles, such as clinical trials). Subsequently, a number of 137 full studies were screened, and 132 were eliminated: 59 studies regarding other ocular pathologies, not open-angle glaucoma (such as cataract, diabetic retinopathy, age-related macular degeneration, Leber’s optic neuropathy), 49 regarding systemic disease (including dermatological and oncological disease), 1 article related to healthy subjects and 1 article related to the effects of normal aging. Interestingly, some eliminated manuscripts involved mainly mitochondrial dysfunction or oxidative stress, processes in which nicotinamide adenine dinucleotide is highly involved in. The final five articles were: three retrieved from PubMed and two retrieved from Web of Science.

### 3.2. Study Characteristics

After searching the databases and applying the conditions mentioned beforehand, five case–control studies comparing dietary niacin intake in glaucoma and healthy subjects were identified, performed in a South Korean population [25,26] and in a United States of America population [27,28,29] (See Table 3). Risk of bias was assessed using the NOS scale (see Table 3, column Bias assessment (NOS scale)). All studies were awarded nine stars based on the NOS scale: four stars in terms of selection (case and definition adequate, all cases and controls selected from the community), two stars in terms of comparability (all results controlled for age and several other confounding factors—sex, race, educational level), three stars for exposure (all interviews in the studies were blinded to the case/control status of the subject, exposure and outcome were measured identically in the whole cohort, response rate was reported globally and for cases and controls).

### 3.3. The Association Between Daily Niacin Consumption and Glaucoma Using Self-Diagnosis as Diagnostic Criteria

Self-diagnosis has been used as a criterion to categorise glaucoma in two studies, Hou et al. and Taechameekietichai et al. [27,29]. The amount of niacin ingested daily differed significantly: while patients with self-reported glaucoma had a calculated intake of 21.14 mg/day, patients without glaucoma had an intake of 24.35 mg/day (*p* < 0.001) [27]. A small additional risk of glaucoma has been identified per mg less of niacin intake; however, the difference is non-significant in multivariate analysis if taken into account the age, sex, race, educational level, smoking, diabetes, cataract surgery, daily total energy, caffeine intake and the intake of other vitamins from the B group [29].

Both studies classified participants into quartiles based on the daily niacin intake (see Table 4). The odds ratio of being diagnosed with glaucoma was 0.48 for patients with a daily niacin intake of over 28.43 mg, compared to those with a daily intake of niacin of 15.01–20.72 mg [29], and 0.57 for patients with a daily niacin intake of over 28.23 mg, compared to those with a daily intake of less than 15.33 mg [27]. However, these differences become statistically non-significant after adjusting for factors such as age, sex, race, educational level, total energy intake and diabetes diagnosis [27,29], as well as additional factors including smoking, cataract surgery, caffeine intake and the daily intake of other B vitamins [29].

### 3.4. The Association Between Daily Niacin Consumption and Glaucoma Using Retinal Imaging as Glaucoma Diagnosis of Patients

Two studies, comprising 180 patients with glaucoma and 9737 patients without glaucoma, were included in this meta-analysis [27,29]. No heterogeneity was observed between the studies (*p*-value > 0.05 and I^2^ < 80% for all four quartile analyses), allowing us to apply a fixed-effects model. A significant decrease in the odds ratio for glaucoma diagnosed via retinal imaging was observed from the first to the fourth quartile. Notably, there were significant differences in Q1 niacin consumption between patients with and without glaucoma (*p*-value = 0.004), as illustrated in Figure 2A, with a higher prevalence of glaucoma in the Q1 niacin consumption group (OR = 1.59, 95% CI 1.16–2.17). In contrast, no significant difference was found in Q2 niacin consumption between patients with and without glaucoma (*p*-value = 0.38), as shown in Figure 2B, indicating similar glaucoma prevalence in this group (OR = 1.16, 95% CI 0.83–1.62). Similarly, no significant difference was observed in Q3 niacin consumption between patients with and without glaucoma (*p*-value = 0.15), as shown in Figure 2C, indicating comparable prevalence of glaucoma in this group (OR = 0.76, 95% CI 0.53–1.10). In the pooled analysis, a significantly higher proportion of patients with high niacin consumption was found in the group without glaucoma compared to those with glaucoma (*p*-value = 0.02; OR = 0.63, 95% CI 0.43–0.94), as illustrated in Figure 2D. Due to the limited number of studies included in our meta-analysis, a funnel plot was not performed to assess potential asymmetry and bias.

Additionally, Lee SY (2023) has analysed daily niacin intake as a continuous variable in a multivariate analysis, proving that 1 mg of additional daily niacin intake is associated with 6% lower odds of glaucoma diagnosis through retinal imaging [adjusted odds ratio (OR) = 0.94, 95% CI = 0.90, 0.98] [28].

### 3.5. The Association Between Daily Niacin Consumption and Glaucoma Using ISGEO Criteria for Glaucoma Diagnosis of Patients

Two studies, comprising 886 patients with glaucoma and 20,423 patients without glaucoma, were included in this meta-analysis [27,29]. No heterogeneity was observed between the studies (*p*-value > 0.05 and I^2^ < 80% for all four quartile analyses), allowing us to apply a fixed-effects model. A significant decrease in the odds ratio for glaucoma diagnosed via ISGEO criteria was observed from the first to the fourth quartile. Notably, there were significant differences in Q1 niacin consumption between patients with and without glaucoma (*p*-value < 0.0001), as illustrated in Figure 3A, with a higher prevalence of glaucoma in the Q1 niacin consumption group (OR = 1.50, 95% CI 1.30–1.73). Significant differences were found in Q2 niacin consumption between patients with and without glaucoma (*p*-value = 0.03), as shown in Figure 3B, indicating low glaucoma prevalence in this group (OR = 1.16, 95% CI 0.83–1.62). Similarly, no significant difference was observed in Q3 niacin consumption between patients with and without glaucoma (*p*-value = 0.15), as shown in Figure 3C, indicating a comparable prevalence of glaucoma in this group (OR = 0.83, 95% CI 0.70–0.98). In the pooled analysis, a significantly higher proportion of patients with high niacin consumption was found in the group without glaucoma compared to those with glaucoma (*p*-value < 0.00001; OR = 0.66, 95% CI 0.55–0.79), as illustrated in Figure 3D. Due to the limited number of studies included in our meta-analysis, a funnel plot was not performed to assess potential asymmetry and bias.

Data from Hou et al. (2024) regarding glaucoma prevalence as defined by ISGEO criteria were not available for meta-analysis; however, these data are summarised in Table 5. Multivariate analysis showed that 1 mg of additional daily niacin intake was associated with 6% lower odds of glaucoma (adjusted odds ratio (OR) = 0.94, 95% CI = 0.89, 0.99) [29]. However, no significant differences were found between quartiles while considering confounding variables [29].

Lastly, Lee JY (2020) further stratified glaucoma odds ratio on gender and body mass index (BMI). Overall, the daily niacin intake did not differ significantly between patients with glaucoma and healthy subjects. In women, patients with glaucoma had significantly lower daily niacin intake (12.45 mg/day, versus 14.97 mg/day, *p* value 0.003), correcting for age, diabetes, hypertension, high-density lipoprotein cholesterol, intraocular pressure, smoking and drinking alcohol. If stratifying women based on BMI, there was lower niacin intake in glaucoma (*p* value 0.037) only in women with BMI between 18.5 and 23 when considering the aforementioned confounders. No significant differences were found in men [26].

## 4. Discussion

Viewed broadly, all these studies have shown a significant interaction between niacin intake and glaucoma prevalence. Based on NHANES data from 2005 and 2008, we investigated the relationship between dietary intake and glaucoma. Through our meta-analysis, we found that the prevalence of glaucoma is higher in populations with a lower daily dietary intake of niacin. This micronutrient is found mostly in meat (poultry and fish), legumes, nuts and seeds [32]. It is notable, however, that the precise quantity of niacin in the four quartiles of intake was different between studies: in the South Korean population, Q1 corresponded to an intake of <10.15 mg/day [20], while in the United States populations, the lowest consumption was higher (under approximately 15 mg/day [27,29] or under 19–20 mg/day [28,29]). Similarly, the highest quartile of intake was different between studies. This suggests that there is an optimum daily intake of niacin which probably is the most efficient in supporting ocular metabolism; however, more studies are needed to determine it. There have been several methods of glaucoma definition in these studies (self-diagnosis, retinal imaging or ISGEO criteria); however, the odds ratio between quartiles was comparable between self-diagnosis and ISGEO criteria (OR between 0.57 and 0.45), with somewhat better “protective” effects of niacin when defining glaucoma based on retinal imaging (OR 0.36–0.16).

When accounting for niacin as a continuous variable, more modest results were found, with there being either a small or nonsignificant effect on glaucoma prevalence per mg of daily niacin intake [28].

In multivariate analysis, when considering variables such as race, gender, educational level and other nutritional characteristics such as other B vitamins intake, the difference in glaucoma prevalence became insignificant, with only Jung et al. (2018) maintaining a statistically significant OR between quartiles in multivariate analysis [25]. Interestingly, Lee et al. (2020) investigated the differences in niacin consumption while taking into account the gender and the body mass index and found a significant difference in average niacin intake between glaucoma and normal patients only in women with a BMI between 18.5 and 23 [26]. This supports the idea that, as niacin is involved in multiple metabolic processes in the body, there are many factors which modulate the interaction between dietary niacin and glaucoma prevalence.

In addition to dietary intake, there are some data in the literature evidencing a lower serum niacinamide concentration in patients with glaucoma. A quantitative method combining liquid chromatography and mass spectrometry showed that while patients with primary open-angle glaucoma had a plasma nicotinamide concentration of 0.14 μM, healthy controls had a concentration of 0.19 μM [33]. Further, a metabolomics study confirmed that nicotinamide was significantly lowered in glaucoma patients [34,35].

There are promising data in animal models that niacinamide may prove a valuable intervention in glaucoma [21]. Clinical trials have been performed including human subjects, with conflicting results. A double-blind, randomised clinical trial had a cross-over design and evaluated the effect of oral nicotinamide on electroretinography (ERG) and perimetry in patients with glaucoma in comparison with a placebo [36]. Some electroretinography parameters were significantly improved in the study group—the amplitude of the photopic negative response wave (PhNR), with a *p* value of 0.03; and the Vmax ratio (the ratio between the amplitudes of PhNR and the b-wave), with a *p* value of 0.02—both parameters reflecting retinal ganglion cell function [36]. However, no improvement was found in global indices regarding perimetry, intraocular pressure or retinal nerve fibre layer [36].

Another phase 2, double-blind, randomised study tested the visual field of glaucoma patients receiving either placebo or a combination of nicotinamide and pyruvate (a glycolysis product, which was proven in animal models to decline in the retinal tissue concomitant with an IOP increase and to perform a neuroprotective effect when supplemented orally [37]). Several improvements were noted: the number of visual field test locations which improved following the treatment was significantly higher in the study group compared with the control group, and there was a significantly higher rate of improvement in pattern standard deviation (PSD), but there was no improvement regarding the mean deviation (MD) or Visual Field Index(VFI) [38].

Lastly, an OCT investigation of blood perfusion density in the optic nerve and macular areas in glaucoma patients showed a significant improvement following oral niacinamide supplementation, however only in the temporal quadrant of the macula, with better results in more advanced cases of glaucoma [39]. At the time of this writing, there are several randomised controlled trials registered involving niacinamide in glaucoma, therefore promising the potential for robust data in the near future [40,41,42]. Interestingly, niacin supplementation in another ophthalmic disease, age-related macular degeneration (AMD), has shown benefits related to choroidal blood flow and retinal arterial dilation, suggesting a two-fold benefit of niacinamide, both against oxidative and ischemic damage [43].

We acknowledge the importance of addressing potential biases, particularly those inherent to non-randomised studies, such as recall bias in dietary assessments. We recognise that non-randomised studies are prone to confounders and biases, and we reviewed each study’s approach to confounding factors. The systematic review has the limitation that there are differences in confounders included in the multivariate analysis, in the definition of glaucoma and in the quantity of daily niacin corresponding to the lowest and the highest quartiles between studies. Furthermore, the meta-analysis of the data is limited by the limited access to data from the case–control studies. We included only two studies in all meta-analyses, and despite the high number of participants, there are some limitations: limited generalizability (the findings might not be generalizable to a broader population especial the geographic location of the two studies is USA) and overrepresentation of a single study (we observed one study had a stronger effect size and it may have dominated the pooled result, making the meta-analysis essentially a reflection of the findings from that one study, skewing the findings toward some outcomes and giving false positive results).

Niacinamide was widely studied in other indications and has shown a good safety profile and was well tolerated [44]. The systematic review by Gasperi et al. found that dietary niacin may offer protection against Alzheimer’s disease, Parkinson’s disease and age-related cognitive decline, as well as other neurological diseases such as ischemic and traumatic injuries, headache and psychiatric disorders [45]. As more data accumulate from randomised controlled trials and an optimal daily intake is established, niacinamide may prove a valuable addition to glaucoma medication.

## 5. Conclusions

In conclusion, this meta-analysis showed that the daily dietary intake of niacin is significantly lower in patients with glaucoma compared to the general population. Given the different average daily intakes of niacin in these populations, different glaucoma definitions and several confounding variables which weaken the associations, large sample, standardised randomised controlled trials are needed to confirm the potential benefits of niacin in glaucoma.

## Figures and Tables

**Figure 1 nutrients-16-03604-f001:**
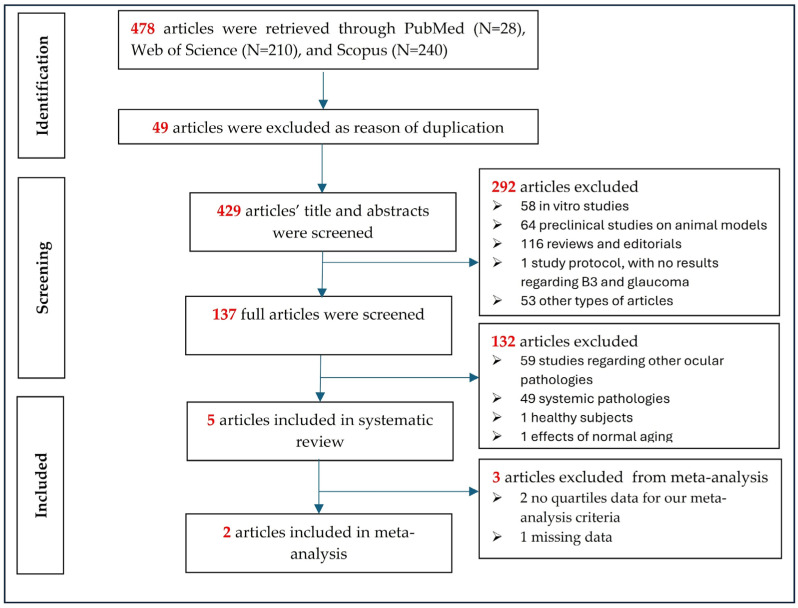
PRISMA flow chart of the study selection.

**Figure 2 nutrients-16-03604-f002:**
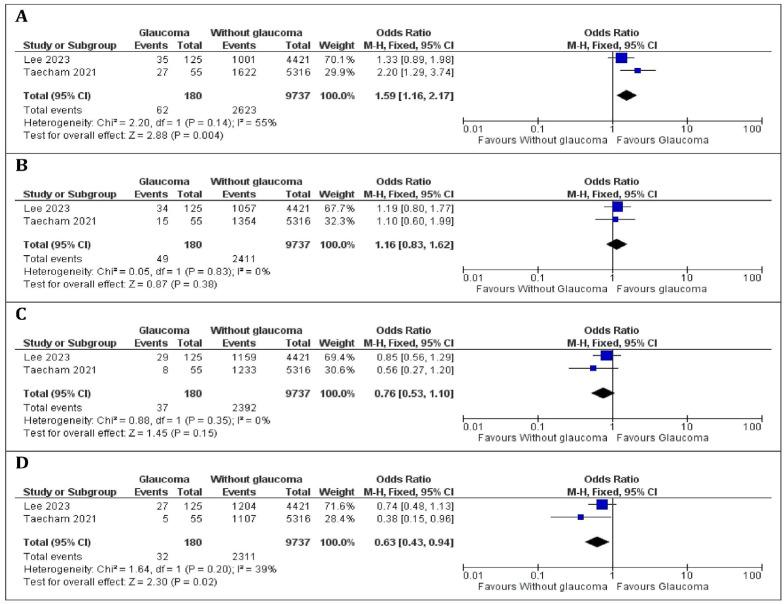
Forest plot of the studies included in the meta-analysis of the association between daily niacin consumption and glaucoma using retinal imaging as glaucoma diagnosis of patients. (**A**) Q1 quartile; (**B**) Q2 quartile; (**C**) Q3 quartile; (**D**) Q4 quartile [27,28].

**Figure 3 nutrients-16-03604-f003:**
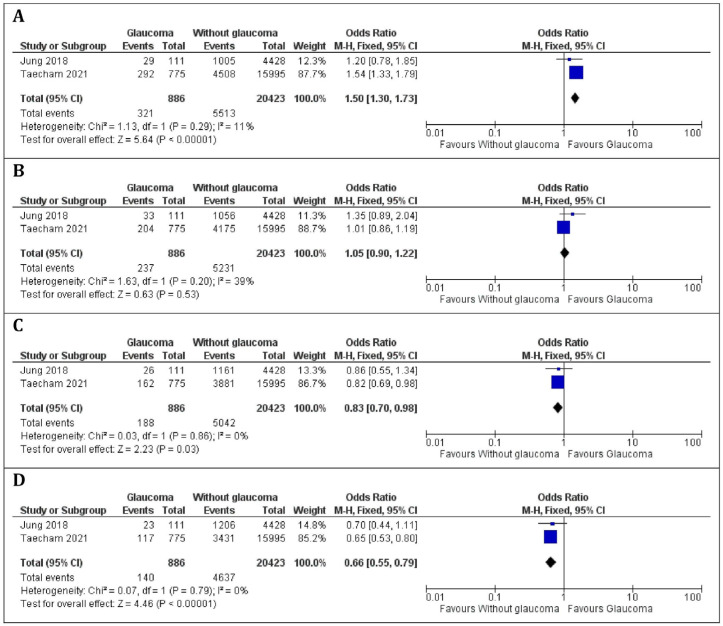
Forest plot of the studies included in the meta-analysis of the association between daily niacin consumption and glaucoma using ISGEO criteria for glaucoma diagnosis of patients. (**A**) Q1 quartile; (**B**) Q2 quartile; (**C**) Q3 quartile; (**D**) Q4 quartile [25,27].

**Table 1 nutrients-16-03604-t001:** The search methodology: summary of keyword combinations searched.

Database	Keywords Combination
Pubmed	((glaucoma[Title]) OR (glaucomatous neuropathy[Title]) OR (primary open angle glaucoma[Title])) AND ((vitamin B3[Title]) OR (nicotinic acid[Title]) OR (nicotinamide[Title]) OR (niacin[Title]) OR (niacinamide[Title]))
Scopus	(TITLE-ABS-KEY (glaucoma) OR TITLE-ABS-KEY (glaucomatous AND neuropathy) OR TITLE-ABS-KEY (primary AND open AND angle AND glaucoma)) AND (TITLE-ABS-KEY (nicotinamide) OR TITLE-ABS-KEY (niacinamide) OR TITLE-ABS-KEY (niacin) OR TITLE-ABS-KEY (vitamin AND b3))
Web of Science	((TS = (glaucoma) OR TS = (glaucomatous neuropathy) OR TS = (primary open angle glaucoma))) AND (TS = (vitamin B3) OR TS = (nicotinic acid) OR TS = (nicotinamide) OR TS = (niacin))

**Table 2 nutrients-16-03604-t002:** The search methodology: inclusion and exclusion criteria.

Studies’ Inclusion Criteria	Studies’ Exclusion Criteria
Study methodology: case–control studies and cohort studies.	Other study methodologies: in vitro studies, animal model studies, reviews, systematic reviews and meta-analyses, editorials, manuscript detailing only a study protocol with no projected results.
At least one glaucoma study group.	No glaucoma study group (only other ocular disease, such as optic neuropathies).
Data on daily intake of vitamin B3/niacin/niacinamide.	No data on daily intake of vitamin B3/niacin/niacinamide.
	Data on other nutrients’ daily dose or supplementation.

**Table 3 nutrients-16-03604-t003:** Summary table of studies included in this systematic review.

ID		Population					Niacin Intake		Glaucoma Outcome			Bias Assessment (NOS Scale)
Author (Year)	Study Design	Country	Sample Size	Source of Participants	Age, Mean	Female, %	4 Quartiles	Continuous Variable	Self—Reported	Retinal Imaging	ISGEO Criteria	
Jung KI (2018) [25]	Case–control	South Korea	16,770	Community	55.45	58.84%	✓				✓	9 ✵
Lee JY (2020) [26]	Case–control	South Korea	6742	Community	43.21	53.16%		✓			✓	9 ✵
Taechameekietichai T (2021) [27]	Case–control	USA	5768	Community	57.5	54.34%	✓	✓	✓	✓	✓	9 ✵
Lee SY (2023) [28]	Case–control	USA	5371	Community	59.35	49.80%	✓	✓		✓		9 ✵
Hou J (2024) [29]	Case–control	USA	5025	Community	56.52	49.19%	✓	✓	✓		✓	9 ✵

✓ signifies the particular way the glaucoma was diagnosed and niacin intake was stratified, ✵ is the standard symbol of the NOS scale, each ✵ signifying a point on the NOS scale.

**Table 4 nutrients-16-03604-t004:** Summary of studies analysing niacin intake as a continuous variable divided in 4 quartiles, and glaucoma as self-diagnosis.

Author (Year)	Study Group (*n*)	Control Group (*n*)	Study Location	Parameters Followed	Quartile as Reference	Univariate Model: OR (95% CI)	*p* Value	Multivariate Model Analysis: OR (95% CI)	*p* Value
Hou J (2024) [29]	331	4694	USA	Daily niacin intake in 4 Quartiles	Q2 (15.01–20.72 mg/day) vs. Q4 (≥28.43 mg/day)	0.48 (0.31, 0.74)	0.001 *	1.04 (0.56, 1.92)	0.899
Taechameekietichai T (2021) [27]	393	5375			Q1 (<15.33 mg/day) vs. Q4 (≥28.23 mg/day)	0.57 (0.37–0.90)	0.018 *	1.39 (0.74–2.58)	0.292

* *p*-value < 0.05.

**Table 5 nutrients-16-03604-t005:** Summary of studies analysing niacin intake as a numerical, continuous variable and as a numerical variable divided into 4 quartiles, and glaucoma diagnosis following ISGEO criteria.

Author (Year)	Size of Control Group	Size of Study Group	Parameters Followed	Results in Univariate Model: OR (95% CI), *p* Value	Results in Multivariate Model: OR (95% CI), *p* Value	Confounders Introduced in Multivariate Model
Hou J (2024) [29]	3156	108	Daily niacin intake as continuous variable	0.97 (0.94, 0.99), 0.011 per 1 mg increase	0.94 (0.89, 0.99), 0.031 per 1 mg increase	age, sex, race, educational level, smoking, diabetes, cataract surgery, daily total energy, caffeine intake and interacted vitamin B
			Daily niacin intake in 4 Quartiles: Q1 (<15.01 mg/day) compared to Q4 (≥28.43 mg/day)	0.45 (0.21, 0.95), 0.036 *	0.36 (0.11, 1.21), 0.099	

* *p*-value < 0.05.

## Data Availability

No new data were created.

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
