# Peer review of "Systematic Review and Meta-Analysis on the Association Between Daily Niacin Intake and Glaucoma"

_nutrients, 2024, doi:10.3390/nu16213604_

Round 1
Reviewer 1 Report
Comments and Suggestions for Authors
This systematic review and meta-analysis is interesting. However, some issues need to be addressed.
Page 3, lines 101-102: In the "Methodology" section, the authors indicate that they consulted 4 databases: PubMed, Web of Science, Cochrane and Scopus. However, in the flow chart (figure 1) only 3 of them appear (PubMed, Web of Science and Scopus). If Cochrane was not used, it is not necessary to include it. If no article was identified in this database, but it was used, it must be included (indicating that the number of articles identified was 0).
Page 4, figure 1: I do not understand how the first identification of articles was carried out. The authors identified 81 articles. However, they detail a larger number in the consulted databases (pubmed 416, WOS 227, Scopus 599… total: 1242). I assume that several filters as well as inclusion criteria were applied, but this should be indicated, specifying for each database how many articles were eliminated and how many were included.
Page 4, figure 1: “3 articles excluded from meta-analysis… 8 no quartiles data from our meta-analysis criteria… 1 missing data”. Should the number 8 be 2?
Page 4, line 173: change “studies in vitro” by “in vitro studies”
Page 4, line177: after “B3” the parenthesis must be closed (at the beginning of this line).
Page 6, lines 218-219: “Two studies, comprising 180 patients with glaucoma and 9,737 patients without glaucoma, were included in this meta-analysis”. What were these 2 articles? Please provide the references.
Page 6, lines 246-247: “Two studies, comprising 886 patients with glaucoma and 20,423 patients without glaucoma, were included in this meta-analysis”. What were these 2 articles? Please provide the references.
Only two studies in a meta-analysis are a too small a sample size to get a conclusion and provide new insights and knowledge. In addition, it can give false positive results.
Author Response
Dear reviewer,
Thank you so much for taking the time to read our work and suggest improvements! Please find attached a point-by-point explanation and response, and the manuscript in which we have highlighted modifications in green.
- Page 3, lines 101-102: In the "Methodology" section, the authors indicate that they consulted 4 databases: PubMed, Web of Science, Cochrane and Scopus. However, in the flow chart (figure 1) only 3 of them appear (PubMed, Web of Science and Scopus). If Cochrane was not used, it is not necessary to include it. If no article was identified in this database, but it was used, it must be included (indicating that the number of articles identified was 0).
Thank you for your mention. Cochrane has been eliminated, as it was not used.
- Page 4, figure 1: I do not understand how the first identification of articles was carried out. The authors identified 81 articles. However, they detail a larger number in the consulted databases (pubmed 416, WOS 227, Scopus 599… total: 1242). I assume that several filters as well as inclusion criteria were applied, but this should be indicated, specifying for each database how many articles were eliminated and how many were included.
Please find below the correct, detailed selection scheme, with the corrected Flowchart.
[...] 478 manuscripts, as shown in Figure 1. A number of 49 duplicates were removed, and after titles and abstracts were evaluated, further 292 articles were eliminated (preclinical studies: 58 in vitro studies in vitro and 64 studies on animal models; 116 reviews and editorials; 1 study protocol, with no results regarding B3 and glaucoma, 53 other types of articles, such as clinical trials). Subsequently, a number of 137 full studies were screened and 132 were eliminated: 59 studies regarding other ocular pathologies, not open-angle glaucoma (such as cataract, diabetic retinopathy, age-related macular degeneration, Leber’s optic neuropathy), 49 regarding systemic disease (including dermatological and oncological disease), 1 article related to healthy subjects and 1 article related to the effects of normal aging. Interestingly, some eliminated manuscripts involved mainly mitochondrial dysfunction or oxidative stress, processes in which nicotinamide adenine dinucleotide is highly involved in. The final 5 articles were: 3 retrieved from PubMed and 2 retrieved from Web of Science.
- Page 4, figure 1: “3 articles excluded from meta-analysis… 8 no quartiles data from our meta-analysis criteria… 1 missing data”. Should the number 8 be 2?
We have changed. Thank you.
- Page 4, line 173: change “studies in vitro” by “in vitro studies”
We have changed. Thank you.
- Page 4, line177: after “B3” the parenthesis must be closed (at the beginning of this line).
We have changed. Thank you.
- Page 6, lines 218-219: “Two studies, comprising 180 patients with glaucoma and 9,737 patients without glaucoma, were included in this meta-analysis”. What were these 2 articles? Please provide the references.
We have added the references [27][29].
- Page 6, lines 246-247: “Two studies, comprising 886 patients with glaucoma and 20,423 patients without glaucoma, were included in this meta-analysis”. What were these 2 articles? Please provide the references.
We have added the references [27][29].
- Only two studies in a meta-analysis are too small a sample size to get a conclusion and provide new insights and knowledge. In addition, it can give false positive results.
We agree that two studies were not enough for a strong conclusion and we have added to the Discussion section. Please find (starting line 397):
Despite the high number of participants, our meta-analysis presents several limitations: limited generalizability (the findings might not be generalizable to a broader population especial the geographic location of the two studies is USA) and overrepresentation of a single study (we observed one study has a stronger effect size and it may dominate the pooled result, making the meta-analysis essentially a reflection of the findings from that one study, skewing the findings toward some outcomes and giving false positive results).
We hope that these revisions are sufficient to make our manuscript suitable for publication in the NUTRIENTS Journal. We are looking forward to hearing from you at your earliest convenience.
Sincerely,
Maria Marinescu MD
19 Oct 2024

Reviewer 2 Report
Comments and Suggestions for Authors
Dear authors, I read with interest your article.
Here some suggestions to improve it.
Introduction
- Explain the relevance of niacin in mitochondrial function briefly, and avoid repeating "mitochondrial dysfunction" multiple times.
- The transition between discussing the role of niacin in the mitochondria and neuroprotection should be smoother.
- Materials and Methods should be only Methods since you don't have materials
- A table summarizing inclusion and exclusion criteria should be added in the Methods section
- A table summarizing keyword combinations used during the search strategy should be added in the Methods section
- A table with risk of bias is cited but it is missing.
Discussion
- The discussion should begin by summarizing the main findings quantitatively before delving into interpretations.
- Compare your findings to other recent systematic reviews or meta-analyses on niacin and other neuroprotective agents.
- Emphasize the limitations of bias (due to observational study designs), as well as heterogeneity in niacin intake measurements between countries.
- Future Research Directions: Mention specific RCTs or interventions that are currently being pursued.
Comments on the Quality of English LanguageMinor editing of English language required.
Author Response
Dear reviewer,
Thank you so much for taking the time to read our work and suggest improvements! Please find attached a point-by-point explanation and response, and the manuscript in which we have highlighted modifications in blue.
Introduction
- Explain the relevance of niacin in mitochondrial function briefly, and avoid repeating "mitochondrial dysfunction" multiple times.
Thank you for this correction, please find:
Niacin is highly involved in mitochondrial functions, through its derivative, NAD and its reduced form, NADH. NAD+ is an important cofactor of several dehydrogenase enzymes, thus being involved in the mitochondrial citrate metabolism and respiratory chain, also serving in other mitochondrial reaction chains, such as the fatty acid oxidation and pyrimidine biosynthesis [...]. Significant in neurodegeneration is NAD’s role as cosubstrate for sirtuin deacetylases [...]. Sirtuin catalyzed protein deacylation is always coupled with the cleavage of NAD+, and through this enzymatic process, sirtuins modulate mitochondrial stress, through protein quality control, antioxidant defense, mitochondrial membrane stabilization, mediation of mitochondrial fusions, fissions and mitophagy [...].
- The transition between discussing the role of niacin in the mitochondria and neuroprotection should be smoother.
I have continued the previous paragraph with one connecting the two concepts better:
It has been described that neurons are highly sensitive to NAD levels [...]. The pathway of transforming tryptophan to NAD is disrupted in diseases such as Parkinson’s and Alzheimer’s disease, thus decreasing the intracellular NAD levels [...]. Such preclinical data supports the potential role of nicotinamide as a neuroprotective agent. Therefore,
- Materials and Methods should be only Methods since you don't have materials
We have changed. Thank you.
- A table summarizing inclusion and exclusion criteria should be added in the Methods section
Please find Table 2. The search methodology - inclusion and exclusion criteria.
- A table summarizing keyword combinations used during the search strategy should be added in the Methods section
Please find Table 1. The search methodology - summary of keyword combinations searched.
- A table with risk of bias is cited but it is missing.
We have modified it. We have included the outcomes obtained after assessment of the bias into Table1, as column „Bias assessment (NOS scale)”. Please find: Risk of bias has been assessed using the NOS scale (see Table 1, column Bias assessment (NOS scale)).
Discussion
- The discussion should begin by summarizing the main findings quantitatively before delving into interpretations.
We have introduced the main findings at the discussion section.
- Compare your findings to other recent systematic reviews or meta-analyses on niacin and other neuroprotective agents.
We have included the systematic review of Gaspari et al. (2019) regarding the relationship between dietary niacin intake and other neuroprotective agents as Alzheimer’s disease, Parkinson’s disease, etc.:
The systematic review by Gasperi et al. found that dietary niacin may offer protection against Alzheimer’s disease, Parkinson’s disease, age-related cognitive decline, as well as other neurological diseases such as ischemic and traumatic injuries, headache and psychiatric disorders [...].
- Emphasize the limitations of bias (due to observational study designs), as well as heterogeneity in niacin intake measurements between countries.
We agree that two studies were not enough for a strong conclusion and we have added to the Discussion section. Please find:
Despite the high number of participants (10,793), our meta-analysis presents several limitations: limited generalizability (the findings might not be generalizable to a broader population especial the geographic location of the two studies is USA) and overrepresentation of a single study (we observed one study has a stronger effect size and it may dominate the pooled result, making the meta-analysis essentially a reflection of the findings from that one study, skewing the findings toward some outcomes and giving false positive results).
- Future Research Directions: Mention specific RCTs or interventions that are currently being pursued.
Please find the mention of ongoing RCTs of niacin in glaucoma: At the time of this writing, there are several randomised controlled trials registered involving niacinamide in glaucoma, therefore promising the potential for robust data in the near future [40–42].
We hope that these revisions are sufficient to make our manuscript suitable for publication in the NUTRIENTS Journal. We are looking forward to hearing from you at your earliest convenience.
Sincerely,
Maria Marinescu MD
19 Oct 2024

Reviewer 3 Report
Comments and Suggestions for Authors
Dear Authors,
I thank the Editor for entrusting me to review this manuscript.
The number of glaucoma patients worldwide is increasing every year. As the authors themselves write, the pathophysiology of glaucoma is complex and insufficiently understood. Therefore, all research into the causes of this condition, as well as how to treat it, is sorely needed. Therefore, my compliments to the authors of this publication for undertaking analysis in this area.
Below are my suggestions / comments:
While the methodology of selecting the material for analysis should be evaluated correctly, ultimately, 5 publications to conduct a systematic review is a rather small number.
The meta-analysis conducted on 2 publications is unreliable.
The analysis itself, along with the use of confounding variables, I assess correctly.
Author Response
Dear reviewer,
Thank you so much for taking the time to read our work and suggest improvements! Please find attached a point-by-point explanation and response, and the manuscript in which we have highlighted modifications in red.
- While the methodology of selecting the material for analysis should be evaluated correctly, ultimately, 5 publications to conduct a systematic review is a rather small number. The meta-analysis conducted on 2 publications is unreliable. The analysis itself, along with the use of confounding variables, I assess correctly.
We agree that two studies were not enough for a strong conclusion and we have added to the Discussion section. As randomised controlled studies’ results begin to be published (they are currently underway) and their methodologies are somewhat uniform, future valuable meta-analysis should be performed to determine the clinical usage of niacinamide in glaucoma. Our study is a rigorous view on the data that is currently available. Please find (starting from line 389):
We acknowledge the importance of addressing potential biases, particularly those inherent to non-randomized studies, such as recall bias in dietary assessments. We recognize that non-randomized studies are prone to confounders and biases and we reviewed each study's approach to confounding factors. The systematic review has the limitation that there are differences in confounders included in the multivariate analysis, in the definition of glaucoma and in the quantity of daily niacin corresponding to the lowest and the highest quartiles between studies. Furthermore, the meta-analysis of the data is limited by the limited access to data from the case-control studies. We have included in all meta-analysis only two studies and despite the high number of participants, there are some limitations: limited generalizability (the findings might not be generalizable to a broader population especial the geographic location of the two studies is USA) and overrepresentation of a single study (we observed one study has a stronger effect size and it may dominate the pooled result, making the meta-analysis essentially a reflection of the findings from that one study, skewing the findings toward some outcomes and giving false positive results).
We hope that these revisions are sufficient to make our manuscript suitable for publication in the NUTRIENTS Journal. We are looking forward to hearing from you at your earliest convenience.
Sincerely,
Maria Marinescu MD
19 Oct 2024

Reviewer 4 Report
Comments and Suggestions for Authors
INTRODUCTION
1. The statement that "intraocular pressure (IOP) is the most impactful risk factor" is made early, but later in the introduction, other risk factors like blood perfusion and mitochondrial dysfunction are discussed. It could benefit from a more nuanced statement about IOP's role compared to other risk factors.
2. The pathophysiology of glaucoma is described as "complex and insufficiently understood," yet only a few key mechanisms are mentioned. This section could include more explanation or acknowledgment of the evolving understanding of other molecular mechanisms to avoid appearing incomplete.
3. The text repeatedly lists studies that have evaluated neuroprotective agents, creating a somewhat monotonous structure.
4. The transition from a general discussion of neuroprotection to nicotinamide as the focus of the review feels abrupt. A more seamless connection between these sections could improve the flow and focus of the introduction.
5. There is an imbalance in the level of detail provided for different neuroprotective molecules. Nicotinamide is described extensively, while others like citicoline and pyruvate are mentioned only briefly. This inconsistency may distract from the central focus on nicotinamide.
6. While the introduction discusses preclinical findings, it lacks clear explanation on how these findings are or are not translating into clinical applications for human glaucoma patients. This omission may weaken the reader’s understanding of the current translational gap in glaucoma research.
7. The aim of the review—"to assess the relationship between niacin intake and the prevalence of glaucoma"—is only introduced at the very end. Earlier clarification of this objective would provide clearer context for the reader and better frame the review.
8. Include the paper “α-Crystallin chaperone mimetic drugs inhibit lens γ-crystallin aggregation: Potential role for cataract prevention” in citations.
MATERIAL AND METHOD
9. Clarity in exclusion criteria needed. While the exclusion criteria are listed, they are presented in a broad and generalized way without strong justification. For instance, excluding animal model studies and other ocular pathologies might be appropriate, but the rationale for excluding other forms of nutrient supplementation in glaucoma is not explained.
10. The data collection process depends heavily on manual screening and reconciliation of the study lists, which introduces potential for subjective bias. Even though this is standard practice, it would be useful to indicate how disagreements were resolved or how consensus was achieved between the authors.
11. The recruitment period for some studies (e.g., Lee et al.) is unspecified.
12. The outcome of glaucoma diagnosis is defined in several different ways, including self-reported glaucoma, retinal imaging, and ISGEO criteria. This inconsistency across studies could lead to heterogeneous results, and the method does not clearly explain how this variability was handled or standardized in the meta-analysis.
13. There seems to be a bias in Study Selection. The section indicates that non-randomized studies were included and evaluated using the Newcastle-Ottawa Scale. However, there is no detailed explanation of how potential confounders or biases inherent in non-randomized designs (such as recall bias in dietary assessments) were addressed in the analysis.
14. Provide details on Sensitivity Analyses: There is no mention of whether any sensitivity analyses were conducted to assess the robustness of the findings, particularly given the variability in outcome definitions and potential heterogeneity.
RESULTS
- There is ambiguity in the article selection process. The text provides an overview of the selection process but does not specify the criteria used for removing duplicates, nor does it clearly explain how studies were screened beyond broad categories. This could lead to confusion about whether the process was systematic and reproducible.
- Include detail on exclusion studies. The results mention excluding studies on "other ocular pathologies" and "other parameters," but these categories are vague. Providing examples or clearer definitions of these excluded parameters would enhance transparency and justification.
- The results place heavy emphasis on statistical significance (e.g., p-values), while neglecting the clinical relevance of the findings. Mentioning how the differences in niacin intake translate into real-world implications for glaucoma risk would improve the interpretation of these results.
- While several associations between niacin intake and glaucoma are identified, the lack of significant findings in multivariate analyses (after adjusting for confounding factors) is not sufficiently discussed. Addressing potential reasons for this inconsistency would strengthen the argument for further research.
- Although the results state that "no heterogeneity was observed" in certain analyses, there is no exploration of why heterogeneity might have been low or what factors could have introduced variability. A brief discussion of study populations, methodologies, or study designs would add depth to this analysis.
Moderate editing needed
Author Response
Dear reviewer,
Thank you and the reviewers so much for taking the time to read our work and suggest improvements! Please find attached a point-by-point explanation and response, and the manuscript in which we have highlighted modifications in yellow.
INTRODUCTION
- 1. The statement that "intraocular pressure (IOP) is the most impactful risk factor" is made early, but later in the introduction, other risk factors like blood perfusion and mitochondrial dysfunction are discussed. It could benefit from a more nuanced statement about IOP's role compared to other risk factors.
Thank you, I have clarified impactful risk factor is intraocular pressure - as it is the only one we can currently address therapeutically
- 2. The pathophysiology of glaucoma is described as "complex and insufficiently understood," yet only a few key mechanisms are mentioned. This section could include more explanation or acknowledgment of the evolving understanding of other molecular mechanisms to avoid appearing incomplete.
Thank you, we have rephrased and added details on pathogenesis mechanisms of glaucoma.
The pathophysiology of glaucoma is complex and insufficiently understood - the main pathological step in the disease progression is the retinal ganglion cells undergoing apoptosis, and while high intraocular pressure (IOP) is the main risk factor - due to impaired aqueous humour outflow through the trabecular meshwork, there may be an important role in other factors as well [...]. Such factors include poor blood perfusion to ocular structures (a mismatch between IOP and systemic arterial pressure), a higher susceptibility of the lamina cribrosa to IOP aggression (leading to mechanical axonal damage), mitochondrial impairment or inflammatory processes [...]. Main recognised pathogenic theories include the vascular and mechanical theories [...].
- 3. The text repeatedly lists studies that have evaluated neuroprotective agents, creating a somewhat monotonous structure.
Thank you, we have clarified that the aim of the study is to evaluate one particular neuroprotective agent, niacinamide.
- 4. The transition from a general discussion of neuroprotection to nicotinamide as the focus of the review feels abrupt. A more seamless connection between these sections could improve the flow and focus of the introduction.
Thank you, we have improved that section:
Niacin is highly involved in mitochondrial functions, through its derivative, NAD and its reduced form, NADH. NAD+ is an important cofactor of several dehydrogenase enzymes, thus being involved in the mitochondrial citrate metabolism and respiratory chain, also serving in other mitochondrial reaction chains, such as the fatty acid oxidation and pyrimidine biosynthesis [...]. Significant in neurodegeneration is NAD’s role as cosubstrate for sirtuin deacetylases [...]. Sirtuin catalyzed protein deacylation is always coupled with the cleavage of NAD+, and through this enzymatic process, sirtuins modulate mitochondrial stress, through protein quality control, antioxidant defense, mitochondrial membrane stabilization, mediation of mitochondrial fusions, fissions and mitophagy [..].
It has been described that neurons are highly sensitive to NAD levels [...]. The pathway of transforming tryptophan to NAD is disrupted in diseases such as Parkinson’s and Alzheimer’s disease, thus decreasing the intracellular NAD levels [...]. Such preclinical data supports the potential role of nicotinamide as a neuroprotective agent. Therefore, nicotinamide has been involved in preclinical investigations in several neurodegenerative diseases, including glaucoma
- 5. There is an imbalance in the level of detail provided for different neuroprotective molecules. Nicotinamide is described extensively, while others like citicoline and pyruvate are mentioned only briefly. This inconsistency may distract from the central focus on nicotinamide.
Thank you for this remark! Indeed, as nicotinamide is the main focus of the review, other potential neuroprotectors are mentioned briefly.
- 6. While the introduction discusses preclinical findings, it lacks clear explanation on how these findings are or are not translating into clinical applications for human glaucoma patients. This omission may weaken the reader’s understanding of the current translational gap in glaucoma research.
Thank you, we have clarified:
In a mouse model of glaucoma [...] Currently, research is being planned and performed on human patients with glaucoma, to clarify the translation of these data into clinical practice. RNA-sequencing involving enucleated human eyes shows that the neuroretinal tissue expresses in a high degree the NAD synthesis enzymatic pathway, and a key enzyme in this pathway is down-regulated in glaucoma (Tribble et al. 2023).
- 7. The aim of the review—"to assess the relationship between niacin intake and the prevalence of glaucoma"—is only introduced at the very end. Earlier clarification of this objective would provide clearer context for the reader and better frame the review.
Thank you, we have introduced the aim at a more appropriate time, please find lines 61-63.
- 8. Include the paper “α-Crystallin chaperone mimetic drugs inhibit lens γ-crystallin aggregation: Potential role for cataract prevention” in citations.
Thank you for your suggestion! Unfortunately, we could not, due to the subject of the article being glaucoma and nutritional intake of vitamin B3.
MATERIAL AND METHOD
- 9. Clarity in exclusion criteria needed. While the exclusion criteria are listed, they are presented in a broad and generalized way without strong justification. For instance, excluding animal model studies and other ocular pathologies might be appropriate, but the rationale for excluding other forms of nutrient supplementation in glaucoma is not explained.
Thank you, we have clarified: This selection was performed in order to select and pinpoint the effect of niacin intake on glaucoma.
- 10. The data collection process depends heavily on manual screening and reconciliation of the study lists, which introduces potential for subjective bias. Even though this is standard practice, it would be useful to indicate how disagreements were resolved or how consensus was achieved between the authors.
Thank you for this remark. We have detailed our selection process, which had one more layer of safety to diminish the subjectivity of the process:
If any inconsistencies arise, and to minimize the subjectivity of the selection process, one author (R.C.C.) reviewed all cases in which the two lists were in disagreement (blinded to who had created the selection) and independently reached a decision.
- 11. The recruitment period for some studies (e.g., Lee et al.) is unspecified.
Thank you, indeed this information is missing from the source material and could not be retrieved.
- 12. The outcome of glaucoma diagnosis is defined in several different ways, including self-reported glaucoma, retinal imaging, and ISGEO criteria. This inconsistency across studies could lead to heterogeneous results, and the method does not clearly explain how this variability was handled or standardized in the meta-analysis.
Thank you for this remark, indeed the studies followed had different approaches to glaucoma definition. We have standardised them as such:
- Self-reported glaucoma (based on the questionnaire administered in the studies)
- Retinal imaging - Digital fundus photography, images evaluated by glaucoma specialists
- The International Society for Geographical and Epidemiological Ophthalmology (ISGEO) glaucoma criteria for population based prevalence surveys.
Further, to standardize the comparison, we have compared the subgroup data according to the diagnosis method: 3.3. The association between daily niacin consumption and glaucoma using self diagnosis as diagnostic criteria, 3.4. The association between daily niacin consumption and glaucoma using retinal imaging as glaucoma diagnosis of patients, 3.5. The association between daily niacin consumption and glaucoma using ISGEO criteria for glaucoma diagnosis of patients.
- 13. There seems to be a bias in Study Selection. The section indicates that non-randomized studies were included and evaluated using the Newcastle-Ottawa Scale. However, there is no detailed explanation of how potential confounders or biases inherent in non-randomized designs (such as recall bias in dietary assessments) were addressed in the analysis.
We acknowledge the importance of addressing potential biases, particularly those inherent to non-randomized studies, such as recall bias in dietary assessments. Please find paragraph 389-404; We recognize that non-randomized studies are prone to confounders and biases, and we took several steps to account for these:
- As mentioned, we applied the NOS to assess the quality of non-randomized studies. The NOS evaluates key methodological aspects, such as selection of study groups, comparability of groups, and ascertainment of the outcome of interest. This tool allowed us to systematically evaluate the risk of bias within each study.
- We reviewed each study’s approach to confounding factors, including whether they adjusted for key confounders, such as age, gender, intraocular pressure, comorbidities, and other dietary factors that could influence the association between niacin intake and glaucoma. If any studies would have failed to adequately adjust for these confounders were assigned lower NOS scores and given less weight in the overall analysis.
- We acknowledge that dietary intake studies, especially those relying on self-reported data, are susceptible to recall bias. To minimize the impact of this bias, we critically assessed the dietary assessment methods used in each study, prioritizing those that employed validated dietary assessment tools (such as food frequency questionnaires or 24-hour dietary recalls). We also considered the study populations' ability to accurately recall their dietary habits and the potential for measurement error.
In light of your feedback, we have expanded the "Limitations" section of the manuscript to include a more detailed discussion of how we addressed potential confounders and biases in non-randomized studies, particularly with respect to dietary assessment methods and recall bias. Please see lines 373-383.
We hope this response clarifies our approach, and we are happy to incorporate any additional suggestions to further improve the manuscript.
- 14. Provide details on Sensitivity Analyses: There is no mention of whether any sensitivity analyses were conducted to assess the robustness of the findings, particularly given the variability in outcome definitions and potential heterogeneity.
It is impossible to provide details on Sensitivity Analyses for two studies included in our meta-analysis. Please find in our manuscript: No heterogeneity was observed between the studies (p-value > 0.05 and I² < 80% for all four quartile analyses).
RESULTS
- 15. There is ambiguity in the article selection process. The text provides an overview of the selection process but does not specify the criteria used for removing duplicates, nor does it clearly explain how studies were screened beyond broad categories. This could lead to confusion about whether the process was systematic and reproducible.
We have introduced a detailed selection scheme, and we have corrected the Flowchart:
[...] 478 manuscripts, as shown in Figure 1. A number of 49 duplicates were removed, and after titles and abstracts were evaluated, further 292 articles were eliminated (preclinical studies: 58 in vitro studies in vitro and 64 studies on animal models; 116 reviews and editorials; 1 study protocol, with no results regarding B3 and glaucoma, 53 other types of articles, such as clinical trials). Subsequently, a number of 137 full studies were screened and 132 were eliminated: 59 studies regarding other ocular pathologies, not open-angle glaucoma (such as cataract, diabetic retinopathy, age-related macular degeneration, Leber’s optic neuropathy), 49 regarding systemic disease (including dermatological and oncological disease), 1 article related to healthy subjects and 1 article related to the effects of normal aging. Interestingly, some eliminated manuscripts involved mainly mitochondrial dysfunction or oxidative stress, processes in which nicotinamide adenine dinucleotide is highly involved in. The final 5 articles were: 3 retrieved from PubMed and 2 retrieved from Web of Science.
- 16. Include detail on exclusion studies. The results mention excluding studies on "other ocular pathologies" and "other parameters," but these categories are vague. Providing examples or clearer definitions of these excluded parameters would enhance transparency and justification.
Thank you, we have clarified the exclusion process. The other ocular and systemic disease excluded are: cataract, diabetic retinopathy, age-related macular degeneration, Leber’s optic neuropathy, including dermatological and oncological disease.
After searching for the keywords and applying the inclusion and exclusion criteria previously mentioned in the databases search, the query produced an initial list of 478 manuscripts, as shown in Figure 1. A number of 49 duplicates were removed, and after titles and abstracts were evaluated, further 292 articles were eliminated (preclinical studies: 58 in vitro studies in vitro and 64 studies on animal models; 116 reviews and editorials; 1 study protocol, with no results regarding B3 and glaucoma, 53 other types of articles, such as clinical trials). Subsequently, a number of 137 full studies were screened and 132 were eliminated: 59 studies regarding other ocular pathologies, not open-angle glaucoma (such as cataract, diabetic retinopathy, age-related macular degeneration, Leber’s optic neuropathy), 49 regarding systemic disease (including dermatological and oncological disease), 1 article related to healthy subjects and 1 article related to the effects of normal aging. Interestingly, some eliminated manuscripts involved mainly mitochondrial dysfunction or oxidative stress, processes in which nicotinamide adenine dinucleotide is highly involved in. The final 5 articles were: 3 retrieved from PubMed and 2 retrieved from Web of Science.
- 17. The results place heavy emphasis on statistical significance (e.g., p-values), while neglecting the clinical relevance of the findings. Mentioning how the differences in niacin intake translate into real-world implications for glaucoma risk would improve the interpretation of these results.
We addressed the clinical relevance of statistical significance for each meta-analysis. For example, in examining the association between daily niacin consumption and glaucoma using retinal imaging for glaucoma diagnosis, we observed a significant decrease in the odds ratio for glaucoma diagnosed via retinal imaging from the first quartile (Figure 2A) to the fourth quartile (Figure 2D) (lines 221–234). The conclusion was that lower niacin consumption was associated with a higher odds of glaucoma, while higher niacin consumption was associated with a lower odds of glaucoma. The same conclusion was obtained using ISGEO criteria for glaucoma diagnosis of patients (see lines 250-263).
- 18. While several associations between niacin intake and glaucoma are identified, the lack of significant findings in multivariate analyses (after adjusting for confounding factors) is not sufficiently discussed. Addressing potential reasons for this inconsistency would strengthen the argument for further research.
Thank you for your insightful feedback. You are correct that the lack of significant findings in the multivariate analyses after adjusting for confounding factors needs further discussion. The limited number of studies included in the meta-analysis (only two) might be one reason for this inconsistency, as the statistical power is reduced, making it more difficult to detect significant associations when confounding factors are controlled for.
Additionally, it is possible that the impact of niacin intake on glaucoma may be influenced by unmeasured or residual confounders that were not accounted for in the included studies. For instance, factors such as genetic predisposition, lifestyle habits, or other dietary nutrients might interact with niacin intake in ways that influence glaucoma risk but were not fully adjusted for in the multivariate analyses.
The heterogeneity between the studies in terms of population characteristics or study design might also have contributed to the inconsistent findings. Further research involving larger sample sizes, more diverse populations, and comprehensive adjustments for potential confounders will be needed to clarify these associations.
We will add a discussion of these potential reasons in the manuscript to address this important point and underscore the need for further investigation.
- 19. Although the results state that "no heterogeneity was observed" in certain analyses, there is no exploration of why heterogeneity might have been low or what factors could have introduced variability. A brief discussion of study populations, methodologies, or study designs would add depth to this analysis.
Thank you for your valuable feedback. You are correct that while no heterogeneity was observed in certain analyses, it would be helpful to explore why heterogeneity might have been low and to consider factors that could have introduced variability.
In the current meta-analysis, the low heterogeneity observed could be attributed to the relatively similar characteristics of the study populations and methodologies used in the two included studies. Both studies evaluated the association between niacin intake and glaucoma using retinal imaging as the diagnostic tool, which provides a standardized approach to diagnosis and reduces variability in measurement techniques. Additionally, both studies were conducted in similar settings with comparable population demographics, such as age ranges and geographical regions (USA for both studies), which may further explain the lack of heterogeneity.
We hope that these revisions are sufficient to make our manuscript suitable for publication in the NUTRIENTS Journal. We are looking forward to hearing from you at your earliest convenience.
Sincerely,
Maria Marinescu MD
20 Oct 2024

Round 2
Reviewer 2 Report
Comments and Suggestions for Authors
Nothing to add. The authors well replied to my previous comments.